# Latin-Square-Based Key Negotiation Protocol for a Group of UAVs

Guangyue Kou *, Guoheng Wei, Zhimin Yuan and Shilei Li

School of Information Security, Naval University of Engineering, Wuhan 430030, China;
wgh7929@aliyun.com (G.W.); yuanzhimin@nudt.edu.cn (Z.Y.); leeshilei@aliyun.com (S.L.)
* Correspondence: m21183902@nue.edu.cn

**Abstract:** Unmanned aerial vehicle mobile ad hoc networks (UAVMANETs) formed by multi-UAV self-assembling networks have rapidly developed and been widely used in many industries in recent years. However, UAVMANETs suffer from the problems of complicated key negotiations and the difficult authentication of members' identities during key negotiations. To address these problems, this paper simplifies the authentication process by introducing a Latin square to improve the process of signature aggregation in the Boneh–Lynn–Shacham (BLS) signature scheme and to aggregate the keys negotiated via the elliptic-curve Diffie–Hellman (ECDH) protocol into new keys. As shown through security analysis and simulations, this scheme improves the efficiency of UAVMANET authentication and key negotiation while satisfying security requirements.

**Keywords:** UAVMANET; multiparty key negotiation; Latin square; BLS protocol



## 1. Introduction

Unmanned aerial vehicles (UAVs) [1] are unmanned aircraft that can be flown autonomously or remotely controlled using wireless channels for communication. The benefits of UAVs include their simple structure, flexible deployment, and low prices. In recent years, with the rapid development and large-scale application of internet-of-things (IoT) technology, the development trend of UAVs has shifted from single UAVs to the cooperative operation of multiple UAVs. UAV mobile ad hoc networks (UAVMANETs) [2] composed of multiple UAVs have become a new type of mobile self-organized networks that are widely used in commercial drone performances, joint search and rescue operations, environmental surveys, military missions, and other applications.

A UAVMANET is a special self-organizing network created by placing clusters of UAVs in open wireless channels [3], through which these UAV clusters can connect autonomously after large-scale deployment. Each node in a UAVMANET has the same status and acts as a temporary relay node while completing its flight mission [4]. The decentralized structure of UAVMANETs ensures greater self-organization, more distributed control, and more dynamic topologies than are found in traditional wireless and wired networks.

However, since the UAV clusters work in insecure open channels [5], UAVMANETs are vulnerable to malicious attackers during the self-assembly process. Such attackers can compromise UAVMANETs by eavesdropping on, jamming, and hijacking message data on the communication links [6]. Key negotiation techniques for establishing secure communication over insecure channels can be applied in UAVMANETs; however, attackers can disguise themselves as legitimate users to obtain session keys illegitimately [7]. Additionally, UAVMANET networking needs to account for the flexibility of the network members. Therefore, there is a need to establish a key agreement scheme that can guarantee the efficient generation of session keys and support any number of UAV group members to ensure the confidentiality, integrity, and availability of data communication. Such a UAVMANET key negotiation protocol should have the following features:

- Extensibility: In key negotiation, any number of UAV group members should be allowed to form a UAVMANET.
- Security: Group members should be secure during the negotiation of group keys, and the final session key information should not be able to be breached by malicious users due to group member key interactions.
- Authenticability: Participating UAVMANETs should be authenticable during key negotiations to prevent man-in-the-middle attacks.

### 1.1. Related Works

UAVMANETs, as a special kind of ad hoc network, have a multiparty key negotiation problem. Solutions to this problem can be divided into two categories: noninteractive key negotiation protocols and interactive key negotiation protocols. Noninteractive key negotiation protocols allow the communicating parties to negotiate the same key in a single key negotiation. After Diffie and Hellman [8] proposed the first noninteractive key negotiation protocol in 1976, many cryptographers attempted to extend this approach to multiple parties, that is, to solve the group key negotiation problem through a single key negotiation. Joux [9] first accomplished the expansion of the Diffie–Hellman (DH) protocol from two to three parties with only one round of communication but did not expand the protocol to more than three members. Garg et al. [10] proposed implementing a multilinear mapping scheme (the GGH scheme) on an ideal lattice using a hierarchical coding system as a solution to the multiparty key negotiation problem. However, this scheme was proven to be unreliable by Hu et al. [11]. Therefore, at present, it is not possible to achieve a noninteractive key negotiation protocol with more than three parties. The research on interactive key negotiation protocols is mainly based on expanding the two-party DH protocol to multiple parties [12,13] by using the DH protocol as the core scheme to form a unified key through the interaction of the protocol participants in multiple communication rounds.

Dutta et al. [14] explored the DH algorithm on a ring structure with forward and backward security but did not support the dynamic joining and leaving of members. Steiner et al. [15] improved the DH protocol by proposing a key agreement approach that can be used for multiple parties and accounts for dynamic group members. However, the number of communication rounds generated in the key update and establishment phase of the protocol is related to the number of group members; as the number of group members increases, establishing group keys becomes more time consuming. Kim [16] formulated a tree-based key management structure to improve the DH protocol and calculated the root node key by cascading the subkeys of the leaf nodes. Compared with other structures, this tree-based key management structure is better suited to the use of the DH protocol in a group environment and can more efficiently reduce the number of node keys [17–20].

Due to the unique mathematical properties of Latin square arrays, they are widely used in the field of communication [21,22]. They can also be applied in key negotiations. Because a given partial Latin square can be uniquely extended to a complete Latin square, a Latin square can be constructed for multiparty key gating. Stones et al. [23] constructed a shared key based on subsecrets using symmetric self-replication. Chum et al. [24] constructed a Latin square key-sharing scheme using hash functions. Shen et al. [25] combined a Latin square scheme with a traditional $(t, n)$-gated key-sharing scheme to optimize machine-to-machine communication by enhancing efficiency and security. We note that in the above applications, the Latin square is load-balancing to adjust the communication model for distributed systems. Boneh et al. [26] proposed using a Latin square to adapt a key negotiation scheme for cloud computing. This protocol supports any number of user members and incorporates key validation and fault tolerance, but its use of multiple mappings is too burdensome for computing on drones.

In the last two years of research on UAV key negotiation, Xia et al. [27] proposed an identity-based elliptic-curve key negotiation scheme to achieve authentication and key negotiation between UAVs and ground stations. However, the proposed system is only

applicable to static UASs with a central node, which is less flexible. Zhang et al. [28] proposed a lightweight authentication and key negotiation protocol for UAVs. The physical unclonable function (PUF) is introduced in the protocol operation, and the authentication and key negotiation can be completed using only hash and heterodyne operations using the characteristics of the PUF, avoiding complex cryptographic operations. However, PUF-based schemes have disadvantages such as complex configuration and the need for specific PUF hardware. Tian et al. [29] proposed a UAV authentication and key negotiation protocol based on the PUF that can communicate across domains. This protocol can communicate across domains before multiple ground stations, but the scheme does not apply to UASs without a central station. Xie et al. [30] managed multiple drone tasks by building a three-tier blockchain. Therefore, this paper proposes using a Latin square to optimize the rounds and process of key negotiation in a self-organizing network of UAVs and designs a set of improved DH protocols to ensure that security and efficiency can be simultaneously addressed in the process of UAV group key negotiation. At the same time, the proposed protocol accounts for the networking characteristics of UAVMANETs and supports a flexible authentication process.

### 1.2. Motivation and Contributions

The main contributions of this paper are as follows:

- We propose a Latin-square-based dynamic-group key negotiation protocol with authentication. Using the strong mathematical and cryptographic properties of Latin squares, we designed the protocol to allow any number of members to form a group and negotiate the session keys through a self-organizing network of group members without the assistance of a central node for key negotiation. Compared with other key negotiation protocols, our protocol has greater decentralization and networking flexibility.

- The proposed protocol is made more efficient by combining a Latin square array with the Boneh–Lynn–Shacham (BLS) signature algorithm. By combining the signature aggregation process with the construction of a Latin square, it is ensured that each round of communication verifies and aggregates the previous round of blocks, achieving a more efficient signature scheme. The traditional protocol requires a communication cost $O(n^2)$, while the proposed protocol has only an $O(n \log n)$ communication cost. The proposed Latin-square-based signature scheme incurs only half the communication overhead of the elliptic curve digital signature algorithm (ECDSA), and this scheme uses curve hashing to manage its time overhead, unlike other schemes.

- The proposed protocol has higher efficiency and less overhead in the key negotiation phase than the traditional protocol. We optimized the broadcast scheme in the traditional key agreement protocol to communicate with specified members in the square; as a result, only an $O(n \log n)$ communication cost is required to complete key negotiation, whereas the traditional key negotiation protocol has a communication cost of $O(n^2)$. Furthermore, in the key agreement stage, we used the elliptic-curve point product algorithm, which incurs less communication overhead. Therefore, the proposed key negotiation protocol is more efficient than the traditional protocol.

### 1.3. Organization

This paper is organized as follows. The first section introduces the concept and main features of UAVMANETs. The second section presents the initial parameters of the protocol along with the mathematical notation used. The third section describes the model used. The fourth section presents the protocol. The fifth and sixth sections analyze the security and key properties of the protocol. The final section summarizes the full text.

## 2. Preliminaries

In this section, we briefly describe the key techniques to be used and clarify their connection to this paper. The symbols that appear in this paper are defined in Table 1.

**Table 1.** Symbolic notations used in the proposed protocol.

| Notation | Description |
|---|---|
| $PID_i$ | UAV identifier |
| $\mathbb{F}_p$ | Domain formed by $\mathbb{G}$ |
| $\mathbb{F}_{p^2}$ | Domain formed by $\mathbb{G}_T$ |
| $\mathbb{G}$ | Additive group |
| $\mathbb{G}_T$ | Multiplicative group |
| $P, Q$ | Prime numbers |
| $\hat{e}$ | Weil pairing on $\mathbb{G} \times \mathbb{G} \to \mathbb{G}_T$ |
| $\mathcal{G}_1$ | The base point of an elliptic curve over a finite field for authentication |
| $\mathcal{G}_2$ | The base point of an elliptic curve over a finite field for key negotiation |
| $Pk_i$ | Drone public key |
| $Sk_i$ | Drone private key |
| $p_i$ | The temporary public key for drones |
| $s_i$ | The temporary private key for drones |
| $M_{t,i}$ | The $t$th negotiated key in the $i$th round |
| $w_{i,j}$ | Shared key of $PID_i$ and $PID_j$ |
| $\kappa$ | Negotiated key |
| $H(s_i)$ | Hash of $s_i$ |
| $Sign_i$ | Signature |

### 2.1. BLS Signature Protocol

Building BLS signatures requires the utilization of curve hashing and the Weil pairing technique.

Curve hashing means that the result of hashing a message corresponds to a point on an elliptic curve, and the construction method is to determine the corresponding points on the elliptic curve for various points whose hash values are plotted on the X coordinate axis.

A Weil pairing is the mapping of two points on a curve to a single number using a special function. Let $E$ be the elliptic curve defined by the equation $y^2 = x^3 + 1$ over $\mathbb{F}_{p^2}$, let $P \in \mathbb{F}_p$ be a point of order $Q$, and let $\mathbb{G}$ be the subgroup of points generated by $P$, where $\mathbb{G}_T$ is a subgroup of $\mathbb{F}_{p^2}$. Then, the map $\varphi(Q)$ is an automorphism of the group of points on the curve $E$. To obtain a nondegenerate map, we define the modified Weil pairing $\hat{e} : \mathbb{G} \times \mathbb{G} \to \mathbb{G}_T$ as follows:

$$\hat{e}(P, Q) = \hat{e}(P, \varphi(Q)) \tag{1}$$

- Bilinearity: $\hat{e}(aP, bQ) = \hat{e}(P, Q)^{ab}$ for all $P, Q \in \mathbb{G}, a, b \in \mathbb{F}_p$.
- Nondegeneracy: If $P \in \mathbb{G}$, then $\hat{e}(P, P)$ is a generator of $\mathbb{G}_T$.
- Computability: There exists an efficient algorithm to compute $\hat{e}(P, Q)$ for all $P, Q \in \mathbb{G}$.

When generating a BLS signature, we first hash the curve of the message and then multiply the coordinate points on the curve obtained from the corresponding curve hash by the private key to obtain the signature. The result is the points on the curve. The signature generation process is shown in Figure 1.

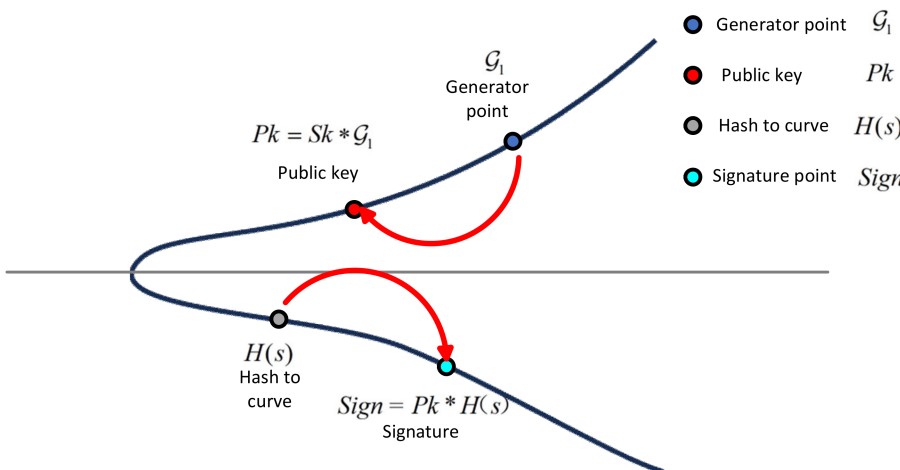

**Figure 1.** BLS signature generation process.

It is necessary to verify that $\hat{e}(Pk, H(s)) = \hat{e}(\mathcal{G}_1, Sign)$ when verifying a signature.

## 2.2. Latin Square

A Latin square is an $n \times n$ square matrix with exactly $n$ different elements in each of the $n$ rows of elements. The pseudocode for the process of Latin square construction is shown in Algorithm 1.

---

**Algorithm 1** Construction of a Latin square

---

**for** x = 1; x ≤ k; x + + **do**
**for** y = 1; y ≤ k; y + + **do**
    $\mathbf{a}_{x,y}$ = (x + y − 1);
    **end for**
**end for**

---

In this paper, using the mathematical properties of a Latin square array, the row elements are used as the communication directions for pairing, and the pairing process forms a new Latin square array to finally aggregate the signature information and key negotiation information. Taking a $4 \times 4$ Latin square array as an example, the specific process is shown in Figure 2.

$$\begin{pmatrix} \mathbf{a}_0 & \mathbf{a}_1 & \mathbf{a}_2 & \mathbf{a}_3 \\ \mathbf{a}_1 & \mathbf{a}_2 & \mathbf{a}_3 & \mathbf{a}_0 \\ \mathbf{a}_2 & \mathbf{a}_3 & \mathbf{a}_0 & \mathbf{a}_1 \\ \mathbf{a}_3 & \mathbf{a}_0 & \mathbf{a}_1 & \mathbf{a}_2 \end{pmatrix} \rightarrow \begin{pmatrix} \mathbf{a}_0 \times \mathbf{a}_1 & \mathbf{a}_1 \times \mathbf{a}_2 & \mathbf{a}_2 \times \mathbf{a}_3 & \mathbf{a}_3 \times \mathbf{a}_0 \\ \mathbf{a}_1 \times \mathbf{a}_2 & \mathbf{a}_2 \times \mathbf{a}_3 & \mathbf{a}_3 \times \mathbf{a}_0 & \mathbf{a}_0 \times \mathbf{a}_1 \\ \mathbf{a}_2 \times \mathbf{a}_3 & \mathbf{a}_3 \times \mathbf{a}_0 & \mathbf{a}_0 \times \mathbf{a}_1 & \mathbf{a}_1 \times \mathbf{a}_2 \\ \mathbf{a}_3 \times \mathbf{a}_0 & \mathbf{a}_0 \times \mathbf{a}_1 & \mathbf{a}_1 \times \mathbf{a}_2 & \mathbf{a}_2 \times \mathbf{a}_3 \end{pmatrix} \rightarrow \begin{pmatrix} \mathbf{a}_0 \times \mathbf{a}_1 \times \mathbf{a}_2 \times \mathbf{a}_3 & \bullet & \bullet & \bullet \\ & \bullet & & \bullet \\ & \bullet & & \bullet \\ & \bullet & & \bullet \end{pmatrix}$$

**Figure 2.** Latin square member aggregation process.

## 2.3. Elliptic-Curve Diffie–Hellman Key Exchange

The elliptic-curve Diffie–Hellman (ECDH) key exchange algorithm is a DH algorithm built on elliptic curves, which uses the dot product operation $(w_i * \mathcal{G}_2) * w_j = (w_j * \mathcal{G}_2) * w_i$ on elliptic curves to negotiate keys. In this paper, the ECDH algorithm is used for UAV key negotiation. The basic units for generating public and private keys and the basic elements for conducting key negotiation are constructed as follows:

- $PID_i$ uses a self-generated random number $s_i$ as a temporary private key, constructs an elliptic curve using the $\mathcal{G}_2$ generated by a ground station (GS), and calculates the public key $p_i$.

- $PID_j$ uses a self-generated random number $s_j$ as a temporary private key, constructs an elliptic curve using the $\mathcal{G}_2$ generated by the GS, and calculates the public key $p_j$.
- $PID_i$ and $PID_j$ exchange their public keys $p_i$ and $p_j$ on an open channel.
- $PID_i$ computes the negotiated key $\kappa = p_j * s_i$.
- $PID_j$ computes the negotiated key $\kappa = p_i * s_j$.
- $PID_i$ and $PID_j$ have the same $\kappa = s_j * \mathcal{G}_2 * s_i$.

In this paper, we complete the key negotiation problem in a group by applying the ECDH algorithm several times in multiple rounds of communication to aggregate the keys, finally ensuring that all members of the group negotiate the same key.

## 3. The Models

### 3.1. System Model

Figure 3 illustrates the communication model of the UAVMANET system. In this system, there are two kinds of entities: a ground station (GS) and UAV nodes [31]. The GS, as a trusted third party in this system, does not participate in key negotiations and is only responsible for providing registration services for members in their first communication. Only members who complete registration can participate in the dynamic activities of the group. The group system consists of several UAVs registered by the GS, communicating through a self-organizing network. When new members need to join, they need to register their unique identifiers (IDs) through the GS. Then, after obtaining the identity information and relevant system parameters provided by the GS, they can interact with other group members to form new session keys. Only UAVs that have registered with the GS, and thus have unique IDs and initial parameters, participate in the authentication and key negotiation process. Therefore, this system is flexible and decentralized.

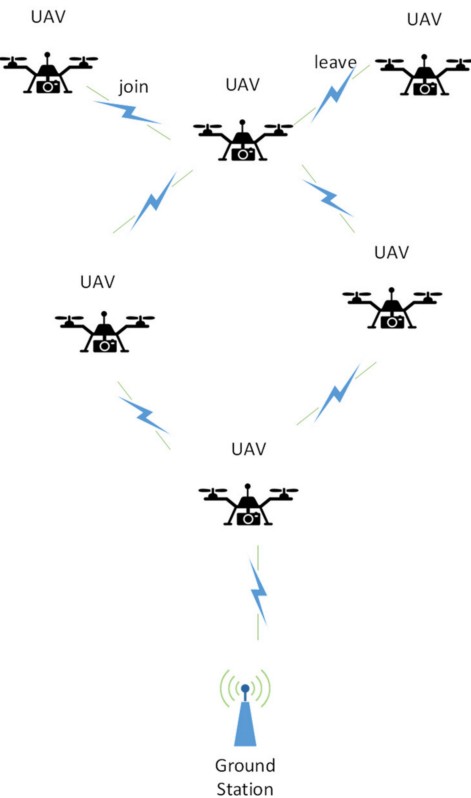

**Figure 3.** The communication model of the UAVMANET system.

### 3.2. Security Model

For this paper, two games, $Game_0$ and $Game_1$, were defined to prove the security of the authentication process and the key agreement process, respectively, of the protocol. The operational model is shown in Figure 4.

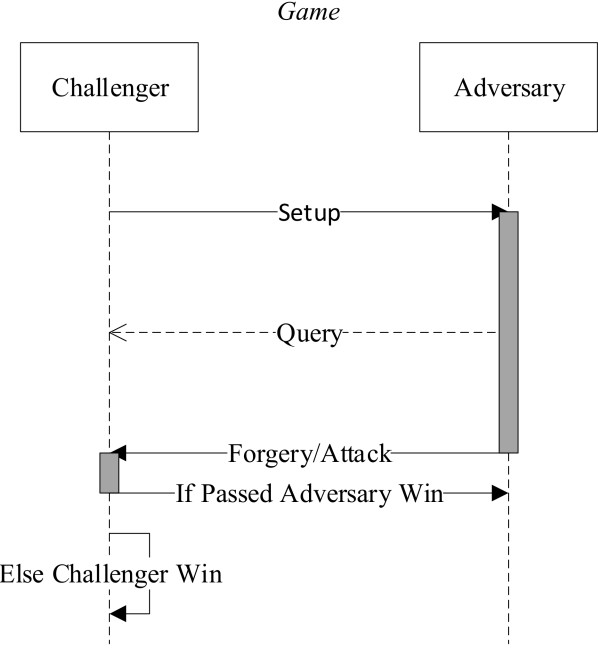

**Figure 4.** Operational flow chart of the security model.

$Game_0$ proves the security of the protocol authentication process. It is a game between an adversary and a challenger under the model of existential unforgeability against chosen-message attacks (EU-CMA) and is designed as follows.

Setup: The challenger $\mathcal{C}$ generates and publishes the initial parameters $\mathbb{PG} = (\mathbb{G}, \mathbb{G}_T, g, p, e)$ by executing the initial phase, which generates $Pk_i$.

Query: The adversary $\mathcal{A}$ selects a drone set $\{PID_1, \ PID_2 \cdots PID_{2^k}\}$ and can repeatedly ask the challenger $\mathcal{C}$ for a public key $Pk_i$ and signature $Sign_i$.

Forgery: When $\mathcal{A}$ finishes querying $\mathcal{C}$, $\mathcal{A}$ forges a signature from the information obtained. If $\mathcal{A}$ forges a correct signature based on the information already queried, then $\mathcal{A}$ wins the game.

$Game_1$ proves the security of the protocol's key negotiation process. It is a game between an adversary $\mathcal{B}$ and a challenger $\mathcal{D}$. The game is designed as follows.

Setup: The challenger $\mathcal{D}$ generates and publishes the initial parameters by executing the initial phase, which generates $Pk_i$.

Query: The adversary $\mathcal{B}$ chooses a drone set $\{PID_1, \ PID_2 \cdots PID_n\}$ and can repeatedly ask the challenger $\mathcal{D}$ for a short-term key $p_i$. The challenger $\mathcal{D}$ replies with the short-term key $p_i$. ($\{PID_1, \ PID_2 \cdots PID_n\} \subset \{PID_1, \ PID_2 \cdots PID_{2^k}\}$, meaning that the adversary $\mathcal{B}$ does not have access to all keys.)

Attack: When $\mathcal{B}$ finishes querying $\mathcal{D}$, the protocol is attacked to recover the negotiated key; if $\mathcal{B}$ can compute the correct key $\kappa$, $\mathcal{B}$ wins the game.

## 4. The Proposed Protocol

This section describes the specific process of a multi-round DH cipher negotiation protocol based on the construction of a Latin square (Figure 5). The protocol is divided into three phases. In the first phase, a Latin square array is constructed for the cluster members for system initialization. Based on the constructed Latin square, the cluster members will select the nodes to perform key negotiations in each round. In the second phase, the cluster

members authenticate their identity information. In the third phase, corresponding cluster members perform key negotiations in accordance with the rules of the Latin square.

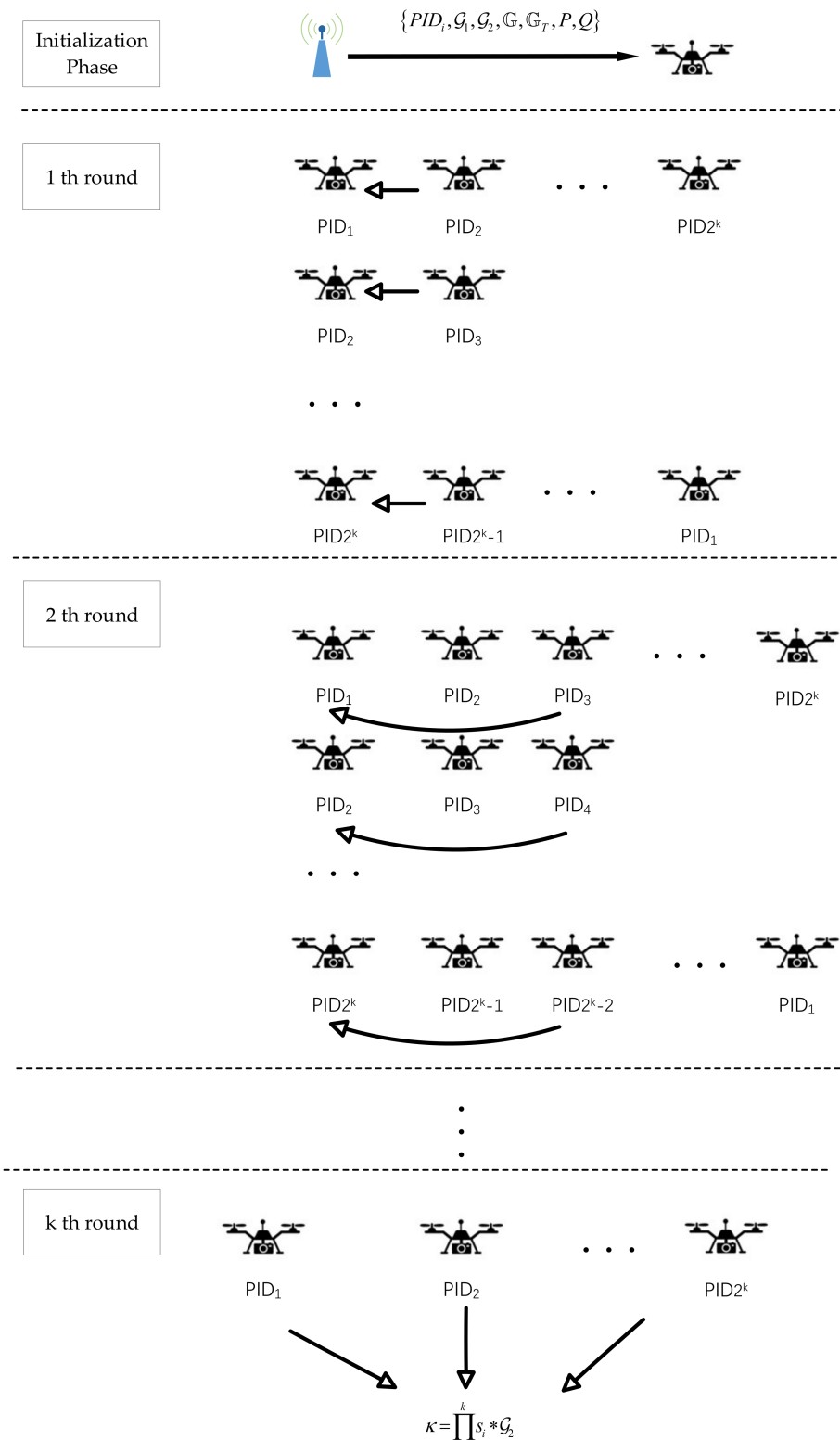

**Figure 5.** Protocol phase diagram.

*4.1. System Initialization*

Before the protocol starts, each UAV obtains a unique identity $PID_i$ by registering with the GS. The base point $\mathcal{G}_1$ of an elliptic curve over a finite field is used to generate the authentication keys $Pk_i$ and $Sk_i$, and the base point $\mathcal{G}_2$ of an elliptic curve over a finite field is used to generate the temporary keys $p_i$ and $s_i$ for negotiation. The GS generates the parameters $\mathbb{PG} = (\mathbb{G}, \mathbb{G}_T, P, Q, \hat{e})$, which are necessary for bilinear mapping, and the GS sends $\{PID_i, \mathcal{G}_1, \mathcal{G}_2, \mathbb{G}, \mathbb{G}_T, P, Q\}$ to each UAV member when it registers with the network.

After a UAV has joined the network, it performs the initialization operation by using the $\{PID_i, \mathcal{G}_1, \mathcal{G}_2, \mathbb{G}, \mathbb{G}_T, P, Q\}$ sent by the GS to generate its long-term key $Pk_i = Sk_i \times \mathcal{G}_1$ and its temporary key $p_i = s_i \times \mathcal{G}_2$, and it calculates $H(s_i)$. The signature $\hat{e}(Pk_i, H(s_i)) = \hat{e}(\mathcal{G}_1, Sign_i)$ is constructed based on the parameters $\mathbb{PG} = (\mathbb{G}, \mathbb{G}_T, P, Q, \hat{e})$.

*4.2. Latin Square Construction*

Suppose that there are three members in a group, denoted by $\mathbf{a_0}$, $\mathbf{a_1}$, and $\mathbf{a_2}$. For this three-member group, the following standard-type Latin square (Latin square in standard form) can be built:

$$\begin{pmatrix} \mathbf{a_0} & \mathbf{a_1} & \mathbf{a_2} \\ \mathbf{a_1} & \mathbf{a_2} & \mathbf{a_0} \\ \mathbf{a_2} & \mathbf{a_0} & \mathbf{a_1} \end{pmatrix}$$

To generalize this Latin square to a generic $k$-order standard-type Latin square model, in the proposed protocol, the total number of Latin square members $n$ is first used to calculate $k = \log_2 n$. If $k$ is not an integer, then to maintain the structure of the protocol, virtual members $2^k - n$ to $2^k$ are added to maintain the structure of the protocol and facilitate the construction of the Latin square.

The generated $k$-order standard Latin square matrix is shown below. For each member of the matrix, in the $x$th row and $y$th column, the element of the matrix is $\mathbf{a_{xy}} = (x + y - 1)$, corresponding to the UAV node $PID_{(x+y)\bmod 2^k}$ in the UAV swarm. By placing the *IDs* of the UAVs into the elements one by one, the constructed square communication matrix model for the UAV swarm can be obtained as shown below.

$$\begin{pmatrix} \mathbf{a_{11}} & \mathbf{a_{12}} & \mathbf{a_{13}} & \cdots & \mathbf{a_{12^k}} \\ \mathbf{a_{21}} & \mathbf{a_{22}} & & & \\ \mathbf{a_{31}} & & \mathbf{a_{33}} & & \cdots \\ \vdots & & & \ddots & \\ \mathbf{a_{2^k1}} & & \cdots & & \mathbf{a_{11}} \end{pmatrix} \Rightarrow \begin{pmatrix} \mathbf{PID_1} & \mathbf{PID_2} & \mathbf{PID_3} & \cdots & \mathbf{PID_{2^k}} \\ \mathbf{PID_2} & \mathbf{PID_3} & & & \\ \mathbf{PID_3} & & \mathbf{PID_5} & & \vdots \\ \vdots & & & \ddots & \\ \mathbf{PID_{2^k}} & & \cdots & & \mathbf{PID_1} \end{pmatrix}$$

Taking a member $PID_1$ as an example, in the first round of communication, $PID_1$ receives a message $Msg_{1,1}$ from $PID_2$ to negotiate the key $M_{1,1}$ after authentication. In the second round of communication, $PID_1$ negotiates the key $M_{2,1}$ with $PID_3$ after authentication, and in the $n$th round, $PID_1$ negotiates the key $M_{n,1}$ with $PID_{2^n}$ after authentication. When $n = k$, indicating the last round of communication, $PID_1$ and $PID_{2^{k-1}+1}$ obtain the final group key $\kappa$. (Note: In this protocol, the default key for virtual members is 1).

After construction through the above process, the UAVs communicate in each round in accordance with the rules of the constructed Latin square, and the two UAVs corresponding to each round interact with each other to aggregate their authentication information and keys and form a new Latin square. Finally, a consistent key is obtained through this aggregation process. The process of signature aggregation confirms the legitimacy of the aggregated key; each member can verify the legality of the whole process, and any illegitimate user will cause errors in the final aggregated signature. Thus, the Latin square construction process ensures efficient authentication and key negotiation. In accordance with the nature of a Latin square, the aggregated information exchanged between the two communicating parties for each round of authentication and key negotiation does not contain duplicate elements, as shown in Figure 6.

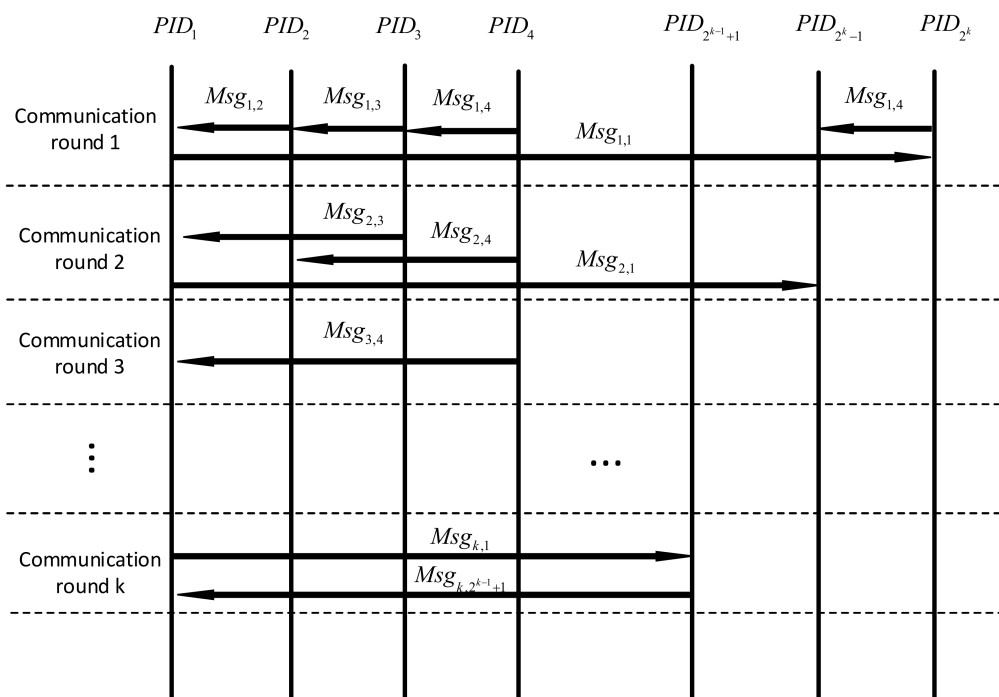

**Figure 6.** Operational process of the proposed protocol.

### 4.3. The Proposed Protocol

The operation of the protocol proposed in this paper is divided into two phases: the authentication phase and the group key negotiation phase.

#### 4.3.1. Authentication Phase

In this phase, the relevant parameters and signatures for authentication are first generated by a single drone. Subsequently, aggregated signatures are formed through interactions with the relevant drones in the corresponding Latin square, and finally, authentication is completed. All participating drones obtain an aggregated signature in this way and can authenticate the identity of any member of the group in any communication round.

In the process of signature aggregation, not only is the information of the participating members authenticated, but the key negotiation process is also recorded, and untrusted individuals can be backtracked by tracing the aggregated blocks. Thus, the aggregated signature results can be used as proof of legitimate participation in the key negotiation process.

Step 1 Generation of public and private keys with individual signatures:

In this phase, each drone $PID_i$ that has registered with the GS generates its own public key $Pk_i$ and private key $Sk_i$ for authentication using the generator $\mathcal{G}_1$ sent by the GS:

$$Pk_i = Sk_i * \mathcal{G}_1 \tag{2}$$

The key $s_i$ of the drone $PID_i$ is signed as follows. First, the hash calculation $H(s_i)$ is performed on $s_i$, and the result is then multiplied by $Sk_i$ to obtain the signature result $Sign_i = Sk_i * H(s_i)$, which is transformed into a point on the elliptic hash curve. The drone sends $Msg_i = \{Pk_i||Sign_i||H(s_i)||\cdots\}$ (the data represented by the ellipses are the second-stage key agreement data) to the corresponding node for authentication.

Step 2 Aggregation of signatures on Latin squares:

A single member generates a signature message by the member communication rules specified by the Latin square constructed as described in the previous section and then starts the first round of communication. During the communication process, the members

participating in each communication round aggregate the signatures from the previous communication round. Eventually, each member can generate a uniform aggregated signature and can verify the signatures from the previous rounds. Based on the difference in the aggregated signatures, the communication round in which an object was sent can be located.

After the drone $PID_i$ receives $Msg_{1,i+1} = \{Pk_{i+1}||Sign_{i+1}||H(s_{i+1})||\cdots\}$ from $PID_{i+1}$ in round 1, it can calculate the aggregated signature using the mathematical properties of an elliptic curve, $Sign = Sign_i + Sign_{i+1}$, while aggregating the key $Pk = Pk_i + Pk_{i+1}$.

Drone $PID_i$ has the aggregated signature $Sign = Sign_{i,1} + Sign_{i,2} + \cdots + Sign_{i,2^n}$ and the aggregated key $Pk = Pk_i + Pk_{i+1} + \cdots Pk_{i+2^n}$ in round $n$ ($1 < n < k$); it receives the following $PID_{i+2^n}$:

$$Msg_{n,i} = \{Pk||Sign||H(s)\} \tag{3}$$

where

$$Pk = Pk_{i+2^n+1} + Pk_{i+2^n+2} + \cdots Pk_{i+2^{n+1}} \tag{4}$$

$$Sign = Sign_{i+2^n+1} + Sign_{i+2^n+2} + \cdots Sign_{i+2^{n+1}} \tag{5}$$

$$H(s) = H(s_{i+2^n+1}) + H(s_{i+2^n+2}) + \cdots H(s_{i+2^{n+1}}) \tag{6}$$

$Msg_{n,i}$ is obtained by using the mathematical properties of elliptic curves to calculate the aggregated signature $Sign = Sign_i + Sign_{i+1} + \cdots + Sign_{i+2^{n+1}}$ while aggregating the key $Pk = Pk_i + Pk_{i+1} + \cdots Pk_{i+2^{n+1}}$.

After the $k$th round of aggregation, $PID_i$ can obtain the aggregated signature $Sign = \sum_{i=1}^{2^k} Sign_i$ and verify signatures with the aggregated public key $Pk = \sum_{i=1}^{2^k} Pk_i$. Similarly, each drone in the cluster can obtain the aggregated signature $Sign$ during the Latin square construction process.

Step 3 Identity verification:

Since an improved BLS signature scheme is used in the signature aggregation process, each round can be considered as a separate block, and performing authentication requires only verifying each block. That is, the following equation should be satisfied: $\hat{e}(\mathcal{G}_1, Sign_i) = \hat{e}(Pk_0, H(s_0)) \times \hat{e}(Pk_1, H(s_1)) \times \ldots \times \hat{e}(Pk_i, H(s_i))$.

After receiving $Msg_{i+1}$ from $PID_{i+1}$ in round one, the drone $PID_i$ verifies the signature using the public key $Pk_{i+1}$ of $PID_{i+1}$:

$$\hat{e}(Pk_{i+1}, H(s_{i+1})) = \hat{e}(Sk_i \times \mathcal{G}_1, H(s_{i+1})) = \hat{e}(\mathcal{G}_1, Sk_i \times H(s_{i+1})) = \hat{e}(\mathcal{G}_1, Sign_i) \tag{7}$$

Drone $PID_i$ uses the aggregated public key $Pk_{i+2^n+1} + Pk_{i+2^n+2} + \cdots Pk_{i+2^{n+1}}$ received from $PID_{(i+2^n)\mathrm{mod}2^k}$ to verify the signature after the first $n$ ($n < k$) rounds when the message $Msg_n$ is received.

Specifically, after receiving $Msg_n$ from $PID_{(i+2^n)\mathrm{mod}2^k}$ in the $n$th ($n < k$) round, the drone $PID_i$ uses the received aggregated public key $Pk_{i+2^n+1} + Pk_{i+2^n+2} + \cdots Pk_{i+2^{n+1}}$ to verify the signature as follows:

$$\begin{aligned}
\hat{e}(Pk, H(s)) &= \hat{e}\big(Pk_{i+2^n+1} + Pk_{i+2^n+2} + \cdots Pk_{i+2^{n+1}}, H(s)\big) \\
&= \hat{e}\big(\mathcal{G}_1 \times (Sk_{i+2^n+1} + Sk_{i+2^n+2} + \cdots Sk_{i+2^{n+1}}), H(s)\big) \\
&= \hat{e}\big(\mathcal{G}_1, Sign_{i+2^n+1} + Sign_{i+2^n+2} + \cdots Sign_{i+2^{n+1}}\big) \\
&= \hat{e}(\mathcal{G}_1, Sign_{i+2^n+1}) \times \hat{e}(\mathcal{G}_1, Sign_{i+2^n+2}) \times \cdots \hat{e}(\mathcal{G}_1, Sign_{i+2^{n+1}})
\end{aligned} \tag{8}$$

Here, the signature block of $PID_{(i+2^n)\mathrm{mod}2^k}$ can be verified only if each signature $Sign_{i+2^n+1}, Sign_{i+2^n+2}, \cdots, Sign_{i+2^{n+1}}$ in the signature block of $PID_{(i+2^n)\mathrm{mod}2^k}$ is valid. In the previous rounds of verification, signature aggregation was performed by other drones, thus saving considerable work.

Finally, after the $k$th round of verification, $PID_i$ verifies the signature $Sign = \sum\limits_{i=1}^{2^k} Sign_i$ using the aggregated public key $Pk = \sum\limits_{i=1}^{2^k} Pk_i$. If the signature is verified, all drones in the entire cluster are legitimate users. Every drone in the cluster can be verified via this method.

4.3.2. Key Negotiation Phase

In this phase, individual drones first generate their public and private keys for negotiation. An aggregated key is then formed by the drones in the corresponding Latin square via the $ECDH$ key negotiation protocol. In the next round, the aggregated key is passed in the same way to form a new aggregated key. Finally, all cluster members can negotiate a common key $\kappa$ without pass-through in the following process.

In the process of key aggregation, the keys are aggregated on an elliptic curve so that members of a group of arbitrary size can negotiate a common key without the participation of the GS in a distributed manner. Thus, the difficult problem of negotiating keys over wireless channels is solved.

Step 1 Generation of public and private keys:

A single drone $PID_i$ generates its own public key $p_i$ and private key $s_i$ for key negotiation using $\mathcal{G}_2$ obtained from the *GS*:

$$p_i = s_i * \mathcal{G}_2 \tag{9}$$

Step 2 Calculation of negotiated and aggregated keys on the Latin square:

In the first round of communication, UAV $PID_i$ receives $p_{i+1}$ from $PID_{i+1}$ and calculates the negotiated key:

$$M_{1,i} = p_{i+1} * s_i = \mathcal{G}_2 * s_{i+1} * s_i \tag{10}$$

In the second round of communication, UAV $PID_i$ receives $M_{1,i+2}$ from $PID_{i+2}$ and calculates the negotiated key:

$$M_{2,i} = M_{1,i} * \mathcal{G}_2 * M_{1,i+2} \tag{11}$$

Drone $PID_i$ forms the aggregated key before the $n$th round ($2 < n < k$):

$$M_{n-1,i} = M_{n-2,i} * \mathcal{G}_2 * M_{n-2,(i+2^{n-1})\bmod 2^k} \tag{12}$$

The following aggregated key is received from $PID_{(i+2^n)\bmod 2^k}$:

$$M_{n-1,(i+2^n)\bmod 2^k} = M_{n-2,(i+2^n)\bmod 2^k} * \mathcal{G}_2 * M_{n-2,(i+2^{n-1}+2^n)\bmod 2^k} \tag{13}$$

The key for this round is calculated as follows:

$$M_{n,i} = M_{n-1,i} * \mathcal{G}_2 * M_{n-1,(i+2^n)\bmod 2^k} \tag{14}$$

After $k-1$ rounds of negotiation, $PID_i$ obtains the key $M_{k-1,i}$, and $PID_{(i+2^{k-1})\bmod 2^k}$ obtains the key $M_{n-1,(i+2^{k-1})\bmod 2^k}$. Therefore, the negotiated shared key $\kappa$ is obtained as follows in the $\kappa$th round:

$$\kappa = M_{k-1,i} * \mathcal{G}_2 * M_{k-1,(i+2^{k-1})\bmod 2^k} \tag{15}$$

By recursively expanding $M_{k-1,i}$ and $M_{n-1,(i+2^{k-1})\bmod 2^k}$ as described above, we can obtain

$$\kappa = \prod_{i=1}^{k} s_i * \mathcal{G}_2 \tag{16}$$

Similarly, all UAVs in the cluster can obtain the shared key by this method.

## 5. Security Analysis

### 5.1. Informal Security Proof

**Theorem 1.** *Each member of the cluster can verify that the negotiated key $\kappa = \prod\limits_{i=1}^{k} s_i * \mathcal{G}_2$ is correct and confidential.*

**Proof.** The negotiated key of UAV cluster member $PID_i$ is $\kappa = M_{k-1,i} * \mathcal{G}_2 * M_{n-1,(i+2^{k-1})\bmod 2^k}$, where $M_{k-1,i} = M_{k-2,i} * \mathcal{G}_2 * M_{k-2,(i+2^{k-1})\bmod 2^k}$, $M_{k-1,(i+2^k)\bmod 2^k} = M_{k-2,(i+2^k)\bmod 2^k} * \mathcal{G}_2 * M_{k-2,(i+2^{k-1}+2^k)\bmod 2^k}$, and so on are calculated recursively downward to obtain $\kappa = \prod\limits_{i=1}^{k} s_i * \mathcal{G}_2$. $\kappa = \prod\limits_{i=1}^{k} s_i * \mathcal{G}_2$ can be transformed into $\kappa = \prod\limits_{i=1}^{k} M_{1,i}$, where computing the private key in each $M_{1,i}$ can be considered equivalent to solving the elliptic curve discrete logarithm problem (ECDLP) puzzle. Therefore, in upward recursion, the aggregated key for each round is also secure. □

**Theorem 2.** *In the protocol authentication phase, each UAV member $PID_i$ in the cluster can form an aggregated public key $Pk = \sum\limits_{i=1}^{2^k} Pk_i$ for the verification of the aggregated signature $Sign = \sum\limits_{i=1}^{2^k} Sign_i$ and can verify that the signature is valid.*

**Proof.** UAV member $PID_i$ in the cluster has formed the following aggregated public key in round $k - 1$:

$$Pk_{i\bmod 2^k} + Pk_{(i+2)\bmod 2^k} + \cdots Pk_{(i+2^{k-1})\bmod 2^k} \tag{17}$$

$PID_i$ receives the following aggregated public key from $PID_{(i+2^{k-1})\bmod 2^k}$:

$$Pk_{(i+2^{k-1}+1)\bmod 2^k} + Pk_{(i+2^{k-1}+2)\bmod 2^k} + \cdots Pk_{(i+2^{k-2}+2^{k-1})\bmod 2^k} \tag{18}$$

The above two aggregated public keys can be summed to obtain $Pk = \sum\limits_{i=1}^{2^k} Pk_i$, and similarly, $Sign = \sum\limits_{i=1}^{2^k} Sign_i$. The signature is verified as follows:

$$
\begin{aligned}
\hat{e}(Pk, H(s)) &= \hat{e}\left(\sum_{i=1}^{2^k} Pk_i, H(s)\right) \\
&= \hat{e}\left(\mathcal{G}_1 \times \sum_{i=1}^{2^k} Sk_i, H(s)\right) \\
&= \hat{e}\left(\mathcal{G}_1, \sum_{i=1}^{2^k} Sign_i\right) \\
&= \sum_{i=1}^{2^k} \hat{e}(\mathcal{G}_1, Sign_i)
\end{aligned}
\tag{19}
$$

This proves the theorem. □

### 5.2. Formal Security Proofs

The formal security proofs are now performed for $Game_0$, $Game_1$ to prove the unforgeability of the protocol with key negotiation against eavesdropping attacks.

$Game_0$:

**Definition 1.** *The Computational Diffie–Hellman (CDH) Problem.*

On the already determined cyclic group $\mathbb{G}$, let $g^a, g^b \subset \mathbb{G}$. Calculating $e(g, g)^{ab}$ is difficult.

Let $Adv^{CDH}(\mathcal{A})$ denote the advantage that A has in trying to break the proposed protocol, defined as follows:

$$Adv^{CDH}(\mathcal{A}) = \Pr[win_A] \tag{20}$$

Let the adversary $\mathcal{A}$ be attempting to forge a signature with a nonnegligible advantage $\sigma$ in solving the CDH problem, expressed as

$$Adv^{CDH}(\mathcal{A}) \geq \sigma \tag{21}$$

Forgery by the adversary $\mathcal{A}$ is considered successful when the following condition is met:

$$\Pr[win_A] \geq \mu \tag{22}$$

According to the security model introduced above, the adversary $\mathcal{A}$ and the challenger $\mathcal{C}$ run $Game_0$ as follows.

First, the challenger $\mathcal{C}$ runs the **Setup** phase to generate the cyclic group $\mathbb{G}$, $g^a, g^b \subset \mathbb{G}$, and its public key $Pk_i$, private key $Sk_i$, and signature $Sign_i$.

Then, the adversary $\mathcal{A}$ performs the **Query** operation, and the challenger $\mathcal{C}$ provides the public key $Pk_i$ of any UAV $PID_i$ in the UAV set $\{PID_1, PID_2 \cdots PID_{2^k}\}$ and the short-term private key hash $H(s_i)$.

When the Query operation has been executed $x$ times, the adversary $\mathcal{A}$ performs the Forgery operation. $\mathcal{A}$ forges a signature based on the obtained data, and the forged aggregated key is $Pk = \sum\limits_{i=1}^{x} Pk_i$ according to the algorithm in the protocol. The decryption algorithm can be used to verify the aggregated signature $Sign = \sum\limits_{i=1}^{x} Sign_i$ with advantage $Adv^{CDH}(\mathcal{A}) \geq \sigma$ on the basis of solving the CDH problem.

$$e\left(\sum_{i=1}^{x} Pk_i, \sum_{i=1}^{x} H(s_i)\right) = e\left(\mathcal{G}_1, \sum_{i=1}^{x} Sign_i\right) \tag{23}$$

To achieve successful forgery, the adversary $\mathcal{A}$ must solve the CDH problem, that is, given $\mathbb{G}, g, g^{Pk_i}, g^{H(s_i)}$, verify $g^{Sign_i \cdot \mathcal{G}_1} = g^{Pk_i \cdot H(s_i)}$. Since there are $2^k$ members in the whole UAV cluster, once the adversary $\mathcal{A}$ has made $x$ queries to obtain $x$ keys, $\mathcal{A}$ still needs to guess $2^k - x$ keys. Let the key length be $d$; then, the probability that the adversary $\mathcal{A}$ wins $Game_0$ is

$$\Pr[win_A] = \frac{1}{2^{(2^k - x) \cdot d}} \cdot Adv^{CDH}(\mathcal{A}) \geq \frac{1}{2^{(2^k - x) \cdot d}} \cdot \sigma \geq \mu \tag{24}$$

If the authentication protocol can be forged, then the advantage in $\Pr[win_A] \geq \mu$ cannot be ignored. If $2^{(2^k - x) \cdot d}$ is also nonnegligible, then the CDH problem has been solved, contradicting Definition 1. Therefore, the authentication part of the protocol is not forgeable.

$Game_1$:

**Definition 2.** *Elliptic Curve Discrete Logarithm Problem (ECDLP).*

Consider the discrete logarithm problem on an elliptic curve with elements $p_i$ on the elliptic curve and base point $\mathcal{G}_2$. Finding $s_i$ under the condition that $p_i = s_i \cdot \mathcal{G}_2$ holds is difficult.

Let $Adv^{ECDLP}(\mathcal{B})$ denote the advantage that $\mathcal{B}$ has in trying to break the proposed protocol, defined as follows:

$$Adv^{ECDLP}(\mathcal{B}) = \Pr[win_{\mathcal{B}}] \tag{25}$$

Let the adversary $\mathcal{B}$ be attempting to forge a signature with a nonnegligible advantage $\sigma$ in solving the ECDLP, to break the ECDLP. This is expressed as

$$Adv^{ECDLP}(\mathcal{B}) \geq \sigma \tag{26}$$

An attack by the adversary $\mathcal{B}$ is considered successful when the following condition is met:

$$\Pr[win_B] \geq \mu \tag{27}$$

According to the security model introduced earlier, the adversary $\mathcal{B}$ and the challenger $\mathcal{D}$ run $Game_1$ as follows.

First, the challenger $\mathcal{D}$ runs the Setup phase, generating the base point $\mathcal{G}_2$, the temporary public key $p_i$, and the temporary private key $s_i$.

Then, the adversary $\mathcal{B}$ performs the Query operation, and the challenger $\mathcal{D}$ provides the temporary public key $p_i$ of any UAV $PID_i$ in the set $\{PID_1, PID_2 \cdots PID_{2^k}\}$.

When the Query operation has been executed $x$ times, the adversary $\mathcal{B}$ performs the Attack operation, attempting to compute the key based on the obtained data. The decryption algorithm is used to solve the ECDLP on the basis of the advantage $Adv^{ECDLP}(\mathcal{B}) \geq \sigma$ in calculating $s_i$. The final negotiated key is obtained as follows by the algorithm in the protocol:

$$\kappa_x = \prod_{i=1}^{x} s_i * \mathcal{G}_2 \tag{28}$$

To achieve successful forgery, the adversary $\mathcal{B}$ must solve the ECDLP, that is, the element $p_i$ and the base point $\mathcal{G}_2$ on the given elliptic curve should identify $s_i$ under the condition that $p_i = s_i \cdot \mathcal{G}_2$. Since the whole UAV cluster has $2^k$ members, once the adversary $\mathcal{B}$ has made $x$ queries to obtain $x$ keys, $\mathcal{B}$ still needs to guess $2^k - x$ keys. Let the key length be $d$; then, the probability of the adversary $\mathcal{B}$ winning $Game_0$ is

$$\Pr[win_B] = \frac{1}{2^{(2^k-x)\cdot d}} \cdot Adv^{ECDLP}(\mathcal{B}) \geq \frac{1}{2^{(2^k-x)\cdot d}} \cdot \sigma \geq \mu \tag{29}$$

If the authentication protocol can be forged, then the advantage in $\Pr[win_B] \geq \mu$ cannot be ignored. If $2^{(2^k-x)\cdot d}$ is also not negligible, this means that the ECDLP has been solved, contradicting Definition 2. Therefore, this protocol can resist eavesdropping attacks.

## 6. Comparative Analysis

This section compares the computational complexity and time overhead, among other characteristics, of the proposed protocol with those of related protocols presented in previous studies [32–34]. The experimental simulations were implemented on a laptop computer with the following specifications: 11th Gen Intel(R) Core(TM) i7-11800H @ 2.30 GHz (16 CPUs). The simulations were implemented using the Python programming language with the PyCryptodome and pypbc libraries, and we chose the class A curve in pypbc to implement bilinear pairing. Table 2 lists the execution times of some operations for comparison with those listed in the literature. For the calculation of the results, the average of 1000 operations was taken.

**Table 2.** The execution times of operations used in the protocol.

| Operation | Symbol | Execution Time (ms) |
|---|---|---|
| Elliptic curve key generation | $t_{ecc}$ | 3.999 |
| Exponentiation | $t_{mi}$ | 3.887 |
| Elliptic curve point addition | $t_{ecc-add}$ | 0.001 |
| Elliptic curve point multiplication | $t_{ecc-mul}$ | 0.431 |
| Bilinear pairing operation | $t_{bp}$ | 4.232 |
| Map-to-point hash operation | $t_{mtp}$ | 4.549 |
| Point addition related to bilinear pairing | $t_{bp-add}$ | 0.094 |
| Multiplication of a scalar with a point based on bilinear pairing | $t_{bp-mul}$ | 1.812 |

In [32], Wei et al. proposed the CL-AAGKA protocol based on group key agreement (GKA). Through identity-based authentication, the key negotiation protocol can be authenticated without certificates. The computational overhead for a single node is $3(n+1)t_{bp} + (2n+1)t_{bp-mul} + 2nt_{ecc-mul}$. With the participation of $n$ nodes, the computational complexity of the system is $O(n^2)$.

In [33], Zhang et al. proposed the IBAAGKA protocol, which is a communication protocol without key escrow based on asymmetric group key agreement (AGKA). Strong unforgeable stateful identity-based batch multi-signatures (IBBMS) were used to ensure that the computational overhead of a single node would be $(n+5)t_{mi} + (5n+1)t_{bp-mul} + 4t_{bp}$; accordingly, the computational complexity of the system is $O(n^2)$ with the participation of $n$ nodes.

In [34], Shen et al. proposed a protocol whose communication model has a reduced computational complexity of $O(n\log n)$ compared to the above two protocols. However, it uses many bilinear pair-based operations for authentication and key negotiation, and its overhead for a single node is $2t_{bp} + 2t_{bp-mul} + (6\log_2 n - 1)t_{mi}$.

The protocol proposed in this paper uses the concept of Latin squares to optimize the communication model, enabling authentication and key negotiation without broadcasting and requiring multicast communication only between nodes. Compared with the above three protocols, the computational complexity is reduced to $O(n\log n)$. In the authentication phase of the protocol, the short BLS-based signature scheme is improved to enable signature aggregation on the Latin square. The mainstream DSA and ECDSA require 320 bits, whereas the BLS short signature algorithm requires only 160 bits. In the key negotiation phase, keys are aggregated using the dot product operation on an elliptic curve, which has a smaller computational overhead than the bilinear pair operation. The overhead for a single node in this scheme is $t_{bp} + t_{mtp} + t_{ecc} + (4\log_2 n - 1)t_{bp-mul} + (2\log_2 n - 1)t_{ecc-mul}$. Table 3 shows a performance comparison of the four protocols.

**Table 3.** Performance comparison of four protocols.

| Protocol | Type of Message Distribution | System Communication Cost | The Computational Cost for Each Node |
|---|---|---|---|
| CL-AAGKA | broadcast | $O(n^2)$ | $3(n+1)t_{bp} + (2n+1)t_{bp-mul} + 2nt_{ecc-mul}$ |
| IBAAGKA | broadcast | $O(n^2)$ | $(n+5)t_{mi} + (5n+1)t_{bp-mul} + 4t_{bp}$ |
| Shen et al.'s protocol | multicast | $O(n\log n)$ | $2t_{bp} + 2t_{bp-mul} + (6\log_2 n - 1)t_{mi}$ |
| Proposed protocol | multicast | $O(n\log n)$ | $t_{bp} + t_{mtp} + t_{ecc} + (4\log_2 n - 1)t_{bp-mul} + (2\log_2 n - 1)t_{ecc-mul}$ |

In this comparison, we used the class A elliptic curve in the pypbc library in Python to calculate the time overhead of each protocol for the cases of 16, 32, 64, and 128 group members. The calculated run times of the four protocols are compared in the form of line

graphs in Figure 7. With 16 members, the protocol proposed in this paper is 2.5 times faster than the Shen et al. protocol, 5.7 times faster than the IBAAGKA protocol, and 12.8 times faster than the CL-AAGKA protocol. In the case of 128 members, the protocol proposed in this paper is 3.9 times faster than the Shen et al. protocol, 15.5 times faster than the IBAAGKA protocol, and 61.5 times faster than the CL-AAGKA protocol.

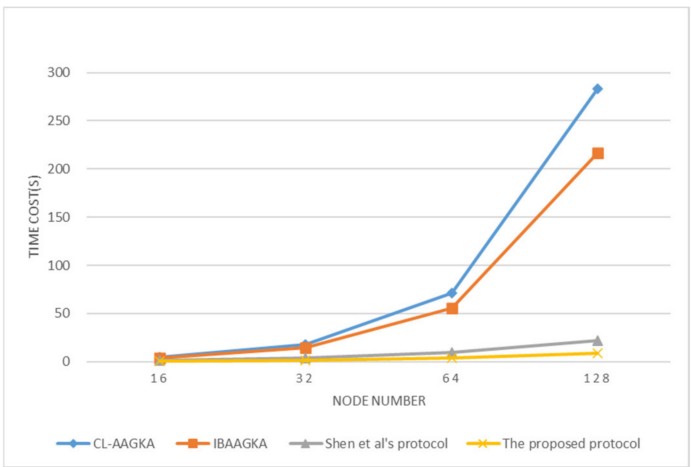

**Figure 7.** Time-cost comparison of the four protocols.

As the number of simulated UAV nodes increases, the run times of the CL-AAGKA and IBAAGKA protocols show exponential growth trends. In comparison, the execution times of the Shen et al. protocol and the protocol proposed in this paper grow more slowly, showing a clear time-overhead advantage. Compared to the Shen et al. protocol, the proposed protocol achieves a lower time overhead by aggregating BLS signatures over a Latin square array with the use of the elliptic-curve dot product operation, which incurs less communication overhead.

## 7. Conclusions

In this paper, we focused on the problems of authentication and key negotiation for a group of UAVs in the context of networking and proposed an aggregated signature-based UAV key negotiation protocol based on the concept of Latin squares. The proposed protocol is well adapted to the characteristics of UAVs communicating via wireless channels and enables the computation of a common key without the participation of a central node in the negotiation process. This paper combined the BLS signature algorithm with the Latin square approach for the first time and proposed a method for completing key negotiation through the aggregation of keys on a Latin square. The proposed protocol is highly flexible and has greater operational efficiency than existing protocols, making it more valuable in UAV environments with limited computing resources.

However, the groups formed by the protocol proposed in this paper need to be studied in more detail when the members join dynamically, and the protocol proposed in this paper needs to be improved and enhanced for situations where the group members change frequently. In the future, we will work on this basis to design a more flexible group key negotiation protocol, focusing on scenarios with frequent changes of group members.

**Author Contributions:** Conceptualization, G.W.; methodology, G.W.; validation, G.K.; formal analysis, G.K. and Z.Y.; investigation, G.K.; writing—original draft preparation, G.K.; writing—review and editing, G.K. and G.W.; supervision, G.W.; project administration, G.W.; and funding acquisition, S.L. All authors have read and agreed to the published version of the manuscript.

**Funding:** This research was funded by National Defense Science and Technology Foundation Enhancement (No. 2019-JCJQ-JJ-042), China.

**Data Availability Statement:** Not applicable.

**Conflicts of Interest:** The authors declare no conflict of interest.

**Abbreviations**

| | |
|---|---|
| UAVs | Unmanned aerial vehicles |
| IoT | Internet of things |
| UAVMANET | Unmanned aerial vehicle mobile ad hoc network |
| BLS | Boneh—Lynn—Shacham signature algorithm |
| DH | Diffie—Hellman key negotiation protocol |
| GGH | Goldreich, Goldwasser, and Halevi mapping scheme |
| GS | Ground station |
| EU-CMA | Existential unforgeability against chosen-message attacks model |
| GKA | Group key agreement |
| AGKA | Asymmetric group key agreement |
| IBBMS | Identity-based batch multi-signatures |

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
