# Peer review of "Latin-Square-Based Key Negotiation Protocol for a Group of UAVs"

_electronics, doi:10.3390/electronics12143131_

Round 1

Reviewer 1 Report

The manuscript reports an interesting study on the transmission of communications between UAVs. The use of UAVs is increasingly frequent and therefore the application of studies in this sector are in great demand and can have a very useful impact on society and the economy.

However, the manuscript needs to be improved as the statistical validity of the data was not analyzed at all. It is not clear whether the reported values are averages or unique data.

In the first case, it is necessary to indicate the characteristics of the data collected to highlight the variability of the values and the significance of the difference between the means.

In the second case, if the values are unique, then the experiment should be repeated to see if randomness had a negligible effect.

Author Response

Dear reviewer::

Thank you for allowing us to submit a revised draft of my manuscript titled “Latin Square-based Key Negotiation Protocol for Group UAVs”. We appreciate the time and effort that you have dedicated to providing your valuable comments and professional advice, which are very helpful for improving our paper. Based on your suggestion and reminionsectedthemeifications resubmitteds to the resubmitted manuscript. Furthermore, We have now worked on both language and readability and have also editethe d manuscript by language s   We really hope that the flow and language level have been substantially improved. Finally, we would like to show the details as follows:

Question 1:In response to h is e q, whethernclear whether the reported vuncannye averagesunclearlearor unique data.

Respyoure:

Thank you for the careful guidance. The protocol we propose in this paper is based on the theoretical validation part of a large project on UAV swarm communication, for the theoretical validation only using simulation rather than real experimental implementation, and we simulated the time overhead of the protocol using Python. For the data in Table 3, we simulated and ran each algorithm 1000 times to take the average of them, for the data in Table 4, we simulated and ran the corresponding protocol 50 times to take the average.

Question 2:Indicate the characteristics of the collected data in response to the proposed statistical validity of the data to highlight the issue of variability of values and the significance of differences between means.

Response:

Thank you for raising this issue. On the one hand, our group studied the causes of errors in the experiments. The source of errors in our simulations is the slight differences in the running results caused by the different CPU occupancy of the computer during each calculation of the protocol overhead in the simulation. The purpose of our simulation is to compare the time overhead of running different protocols whose error sources have little influence on the experimental results. In the actual implementation of the protocol, the factors that hainfluencepac on the time cost are  the influence of the power supply and external tisdifficulttn the is of the UAV, which are difficult to achieve through simulation. We will focus on collecting and analyzing this data as we proceed to implement the protocol on a microcontroller and apply it. Your suggestion will be a guiding suggestion for our subsequent research. On the other hand, after we refer to a large number of related papers studying security protocols, the results of similar studies for protocol overhead simulation mainly corroborate the efficiency of their protocols, which is the weak point of this part of the study.

Reviewer 2 Report

Major:

  *   Please, provide a good English grammar review in the manuscript to remove typos and improve the readability.
  *   The abstract should be rewritten to highlight the motivation, justification, contribution, and results achieved from the experiments.
  *   Section 1.1 must be after related works (section 1.2). Besides, it is important to give more clarifications and comparisons in section 1.2.
  *   In Section 1.2, the first paragraph must be rewritten because it's too confusing, and the words "key negotiation" appears several times.
  *   The variables presented in Section 2 must be explained and what they represent. Insert after a table summarizing them.
  *   More explanations are required in section 4.1.
  *   Insert the numbers in the Equations and call them in the text. Also, call the figures in the text.
  *   More results and discussion must be provided to improve the paper in the Section Comparative Analysis. Compare the obtained results with the literature (also qualitatively).

 Minor:

  *   The abstract contains a few acronyms that the authors should have explained.
  *   The second paragraph of the introduction must contain references to give examples to the reader about the claimed affirmations. The same occurs in the third paragraph.
  *   Correct the font size of the mathematical variables that appear in the manuscript.
  *   Capital letter to "1.1. Related Works"
  *   There is a dot in the wrong place in line 108.
  *   Fix: His section to This section in line 184.
  *   More typos are presented in the manuscript. 

-

Author Response

Please see the attachment(The part in red is the revised part in the original text.)

Reviewer 3 Report

The manuscript provides a clear overview of the paper's objectives, methodology, and findings. It effectively conveys the need for a secure key protocol in UAVMANETs and highlights the use of Latin squares to enhance authentication and key negotiation processes. The mention of security analysis adds credibility to the proposed protocol.

Remarks:

1.      not all the sub-titles begin with capital letter (ex. 1.2. related works, 1.3. organization)- this situation creates confusion.

2.      At line 184 the paragraph begins with “His” probably it should be “This” - this situation creates confusion.

By technical point of view developing such protocol involves solving a series of problems such as:

1.      Whitin the paragraph 4.1 – the authors briefly mentions that the UAV obtains a unique identity (PID) by registering at the GS and uses elliptic curve points to generate authentication and temporary keys. However, the specific steps and algorithms involved in the key generation process are not explained in detail. Clarifying the key generation procedure would enhance understanding - – how did you solve this issue within your research? Please detail into the manuscript.

2.      Whitin the paragraph 4.1 – the authors mention that the GS generates certain parameters necessary for bilinear mapping and sends them to each UAV member during registration. However, the purpose and significance of these parameters are not elaborated upon - providing a clearer explanation of their role in the protocol would improve comprehension.

3.      Whitin the paragraph 4.1 – the authors mention the construction of a Latin square using a standard form but does not provide a clear explanation of how the Latin square is built. It refers to adding virtual members and generating a k-order standard Latin square matrix but does not specify the exact process - providing step-by-step instructions or a more detailed explanation would aid in understanding the Latin square construction.

4.      Whitin the paragraph 4.1 – missing explanation of the purpose of the UAV swarm communication model square matrix: While the text mentions the construction of the square matrix using the UAV IDs, it does not explain the purpose or significance of this matrix in the protocol - Providing further information on how the matrix relates to the communication model or its role in the protocol would be beneficial.

5.      the authors mention the generation of public and private keys for authentication but does not explain how these keys are generated or the specific algorithm used. Additionally, the process of signing the keys is not clearly described. Providing more details on the key generation and signing process would enhance comprehension.

6.      The authors mention the aggregation of signatures during the communication rounds but does not explain the specific mechanism or algorithm used for the aggregation. Clarifying the method of signature aggregation and its purpose in the protocol would improve understanding.

7.      The authors describe the generation of aggregated signatures and keys for each member, but it does not provide information on how these aggregated values are used or their significance in the protocol. Including an explanation of the purpose and role of aggregated signatures and keys would provide a clearer understanding.

8.      Please briefly detail in the conclusions what does it add to the subject area compared with other published material?

Moderate editing of English language required.

Author Response

(The authors gave the same response as above.)

Reviewer 4 Report

The paper proposes a Latin Square-based Key Negotiation along with  Boneh-Lynn-Shacham (BLS) signature algorithm for UAV communication. The topic is hot;  however, the paper has the following points to be considered:

-        The paper is hard to read due to the large number of abbreviations. Please provide a list of abbreviations table by the end of the introduction section to include all symbols.  

-        For the security model, please use the sequence diagram for clarification. 

-        In section 4.3.2 , it is clear how drones generated their own public and private keys.

-        It is better to provide a sequence diagram showing the phases and the operations of the protocol, including the used equations.   

it is ok.

Author Response

(The authors gave the same response as above.)

Round 2

Reviewer 1 Report

Good morning, your comments and suggestions have been taken into account by the authors. Changes have been made to the text. I have no other suggestions to highlight.

Author Response

We have made further revisions to the article. Please see the attachment.

Reviewer 2 Report

The authors provided a significant improvement in this paper. 

My comments are:

  • Remove table 1 (table for acronyms are not necessary);
  • Verify the correct notation for vector multiplication as well as multiplication of scalars;
  • Vector and matrices must be in bold;
  • Correct the Kth term in Fig.5
  • Step 1 (line 373) should be in the next line;
  • In line 481, change the word "literature" to the name of the authors;
  • State the drawbacks of the proposed solution and future improvements in the last section;
  • Are there more works in the literature? Recent ones? Cite them.

Another round of revision must be performed. 

Check the format of the equations and use appropriated mathematical notations for vector, matrices, and multiplication

Reviewer 4 Report

I have no further comments.  The authors addressed my comments.  

NA

Author Response

(The authors gave the same response as above.)

Round 3

Reviewer 2 Report

The authors performed all required suggestions. 

My minor comment is to insert references from 2023 in the manuscript and to perform another grammar check. 

Another round of grammar checks would be beneficial to this paper.
